# A Mathematical Model for the Integrated Optimization of Harvest and Transport Scheduling of Forest Products

**Paulo Amaro Velloso Henriques dos Santos** [1] 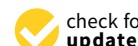**, Arinei Carlos Lindbeck da Silva** [2],
**Julio Eduardo Arce** [3] **and Andrey Lessa Derci Augustynczik** [4],*

[1] Federal Institute of Santa Catarina, Campus Joinville, R. Pavão 1377, 89220-618 Joinville, Brazil;
pauloamaro.ifsc@gmail.com

[2] Department of Mathematics, Federal University of Paraná, ACF Centro Politécnico,
Av. Cel. Francisco H. dos Santos 100, 81531-980 Curitiba, Brazil; arineicls@gmail.com

[3] Department of Forest Sciences, Federal University of Paraná, Campus Jardim Botânico,
Av. Lothário Meissner 632, 80210-170 Curitiba, Brazil; jarce@ufpr.br

[4] Faculty of Environment and Natural Resources, University of Freiburg, Tennenbacherstr 4,
79106 Freiburg im Breisgau, Germany

* Correspondence: andrey.lessa@ife.uni-freiburg.de

**Abstract:** The costs related to forest harvesting and wood transport are key to the economic viability of forest investments. These operations compose a major share of the total cost of wood production and thus need to be conducted in an efficient manner. In this paper, we propose a novel optimization model to tackle this issue and perform the daily and weekly plan of harvesting operations, in order to minimize the costs related to the machinery operation, movement, and wood transportation, subject to demand constraints. Our results show that transportation costs dominate the total cost of these operations. The model proposed is appropriate and can be effectively applied to optimize the operational planning of harvesting activities. Nevertheless, instances with a large number of stands may lead to a substantial increase in the complexity and computational burden. We conclude that operations research techniques can provide a solid basis for decision-making in harvest scheduling problems and increase the efficiency of forest management.

**Keywords:** operational forest planning; forestry transportation; forestry economics; forest harvesting

## 1. Introduction

Operational aspects of forest harvesting are among the main drivers of forest profitability and the sustainability of forest management [1]. Hence, harvesting operations demand adequate and careful planning, in order to optimally allocate the machinery on the field and fulfill forest management goals in an efficient manner. These operations also need to be harmonized with the wood transport from forest areas to demand centers, since transportation costs are responsible for a large share of the total cost of wood supply [2]. Therefore, finding optimized solutions that consider simultaneously the costs related to harvest and transportation is essential to build robust wood supply chains and maintain the profitability of forest investments.

The harmonization of harvesting operations and wood transport, however, is a nontrivial task. This involves costly operations and a large number of decisions to be made in short time spans, under limited resource availability. Operations research offers a unique framework to deal with problems of this nature and has been successfully applied in forest planning problems for more than five decades. The seminal papers dealing with the use of operations research techniques in forestry date back to the

1960s [3–5]. This research line grew in the following decades, with increasing number of applications in forest planning problems (e.g., [6–9]).

Forest planning problems may be distinguished in hierarchical levels, according to the operations and time span involved. Weintraub and Cholaky [10] classify such problems in strategic, tactical, and operational planning. According to Mitchell [11], the strategic planning level typically encompasses long-term decisions that span over more than one rotation period of the forest, e.g., the harvesting schedule, silvicultural operations, and land use. The tactical planning relates to midterm decisions that commonly span from 5 to 10 years, e.g., the planning of road building, the application of environmental constraints, and product usage. Finally, the operational planning deals with short-term decision making (daily and weekly plans), tackling decisions related to the allocation of harvesting machinery and products.

Since the 1990s, a substantial portion of the literature has focused on forest planning, particularly addressing problems at the strategic and tactical levels (e.g., [12–14]). The operational level, which encompasses the activities belonging to the harvest and transport of forest products, has seldom been addressed. A main reason for this gap can be attributed to the fact that operational-level problems are substantially more complex than problems belonging to the other two hierarchical levels. Moreover, it is important to note that optimization processes on this level should deliver solutions in a short computational time, as the operational level deals with monthly, weekly, and daily planning horizons.

According to Rönnqvist et al. [15], the operational level involves multiple decision processes, such as harvesting, loading and distribution of logs, the transport of products harvested in the forest to the customers, and the allocation of machinery needed to perform these tasks. The same authors highlight that the increased access to different sources of information has led to the development of more detailed and complex models and constraints [15].

The scheduling of harvesting machinery across forest stands and the transport of wood to the demand centers are arguably the most important decisions at the operational level and are critical to the maintenance of the wood supply chain. These operations need to be conducted ideally with a minimal cost, while fulfilling the demand center's requirements for product delivery. Additionally, harvesting and transportation operations compose a major share of forest management costs, so a disruption in the wood supply may result in substantial opportunity costs at the demand centers.

Mixed integer linear problems (MILP) have been extensively used for scheduling problems in forestry and other environmental issues. Notably, applications include harvest scheduling, the integration of adjacency relationships, allocation of machinery in the wood industry, and food supply chains [16–19]. The majority of studies applying such approaches, however, focused only on producing harvest schedules that take into account spatial or stand-adjacency constraints.

Despite the large body of literature dealing with forest planning at strategic and tactical levels, models for the operational planning of forest harvesting are still largely missing. Rönnqvist et al. [15] point to the fact that no work could be found to date addressing an integrated solution for the scheduling of the harvest and transport of forest products with a narrow planning horizon, with daily or weekly schedules. Given the importance of these activities to wood supply chain and forest profitability, adequate planning tools are urgently needed. Here, we tackle this issue and propose a novel pure integer linear problem (PILP) model to integrate the allocation of harvesting fronts on the field with the wood transport to the demand centers, hence considering operating, movement and transportation costs. In this sense, we consider the following research questions:

- How can operational aspects of harvest scheduling and wood transportation be integrated into an optimized forest planning model?
- What are the impacts of the model size on the processing times and the total cost of harvesting operations?

To answer the research questions, we developed a PILP model, to minimize the total harvesting costs, including operating, movement, and transportation costs under demand constraints. We

randomly generated 13 problem instances with increasing size and complexity, using real data from pine and eucalypt forest plantations in Southern Brazil. We solved the optimization scenarios by using the software Gurobi (Gurobi Optimization, LLC., Beaverton, OR, USA) – version 8.1.1 (2019), deriving the optimal machinery schedule and assessed the cost components related to the harvesting operations and the transportation to the demand centers.

## 2. Materials and Methods

We build here a novel optimization model for the daily schedule of forest harvesting fronts across forest stands, in order to meet the daily demand for different products (timber assortments) at the demand centers, taking into account transportation costs (see Figure 1). The harvesting equipment, in combination with the skidding equipment, typically composes a harvesting front (e.g., harvester + forwarder + loader or feller buncher + skidder + processor + loader combinations). We aimed to minimize the total cost of harvesting operations, including the harvesting operating cost, depending on the stands' topographic and structural characteristics, the movement cost, i.e., the cost related to moving the machinery between forest stands and the cost related to the transportation of wood from stands to the demand centers. Thereby, we could find an optimal machinery schedule, in terms of where to allocate each harvesting front in each day of the planning horizon, in order to increase the economic efficiency of forest harvesting and transportation.

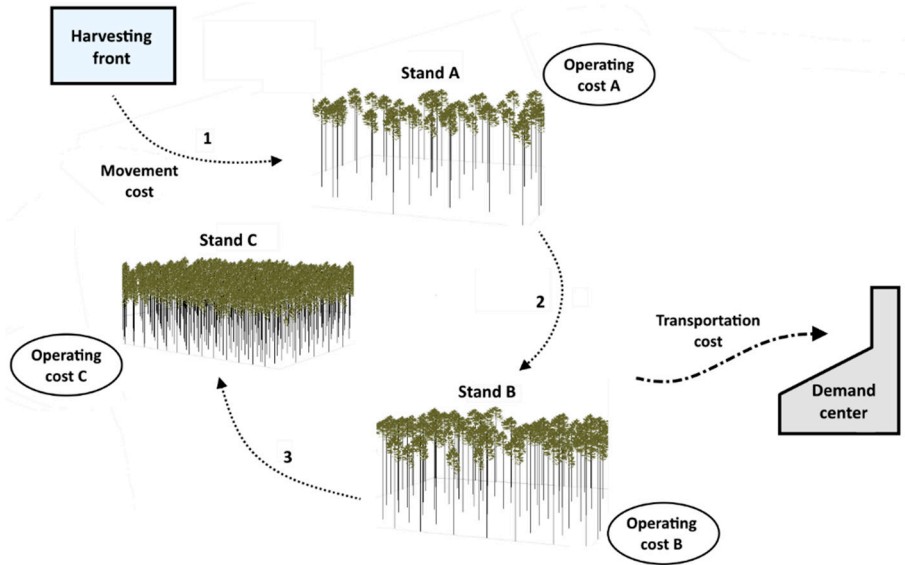

**Figure 1.** Schematic description of the optimization problem.

### 2.1. Model Description

The model we develop here integrates two important and difficult decision problems, namely the schedule of harvesting fronts and the transport of forest products to the demand centers. To address these two problems and all the variables involved in the process, we constructed a PILP model. We describe the model in detail hereafter.

The unidimensional parameters in the model were as follows:

- *T*—Set of available stands;
- *F*—Set of cutting fronts;
- *p*—Set of possible products;
- C—Set of demand centers;
- HP—Planning Horizon;
- $\omega$—Cost of transport for the forest products.

Based on these parameters, indices were defined to be used with the multidimensional parameters and variables in the model. They were as follows:

- $t$—Represents a stand belonging to $T$;
- $f$—Represents a cutting front belonging to $F$;
- $p$—Represents a product belonging to $P$;
- $c$—Represents a demand center belonging to $C$;
- $d$—Represents a harvest day, i.e., a day in which the harvesting front is operating in the stand.
- $\delta$—Represents a day in the Planning Horizon, i.e., a potential candidate for a harvesting front to start harvesting a stand;

The input data of the model was as follows:

- $\mu_f$—Movement cost for front $f$ ($/km);
- $\varphi_f$—Operation cost for front $f$ ($/ha);
- $\tau_{t,f}$—Time needed for front $f$ to harvest stand $t$ (days);
- $Tmax$—Last day in which the harvest can be performed. This term is defined by assuming that a front $f$ would start the harvest of stand $t$ in the last day of the planning horizon, taking into account the time needed for the movement of harvesting front between stands;
- $\rho_{t_1,t_2}$—Distance between stands $t_1$ and $t_2$ (km);
- $\pi_{t,c}$—Distance between stand $t$ and demand center $c$ (km);
- $\lambda_{c,p,\delta}$—Daily demand for product $p$ in demand center $c$ (m$^3$);
- $A_{t,f,\delta,d}$—Area of $t$ that front $f$ will be able to harvest on day $d$ if the harvesting starts at day $\delta$ (ha);
- $\theta_{t,f,\delta,p,d}$—Supply of product $p$ in stand $t$ which the front $f$ will be able to harvest during day $d$ if the harvesting starts on day $\delta$ (m$^3$/day).

The values of $\mu_f$ and $\varphi_f$ are obtained via the definition of the harvesting fronts that will be operating throughout the PH. The productivities of these harvesting fronts were obtained by time and motion studies in the field. With the help of these pieces of information and the area of the stand, the values of $\tau_{t,f}$ and $Tmax$ are defined. The values of $\rho_{t_1,t_2}$ and $\pi_{t,c}$ are obtained in each scenario by the random allocation of stand centroids and the location of the demand center, while $\lambda_{c,p,\delta}$ are related to each demand center and based on real data collected by the students of the Federal University of Paraná. Finally, we are able to derive the harvested area in each stand, depending the starting date and the productivity data, determining the values of $A_{t,f,\delta,d}$ and $\theta_{t,f,\delta,d}$.

Finally, the set of variables is also required for the construction of the model:

- $y_{f,t,\delta}$—Binary, has value 1 if front $f$ starts the harvest of stand $t$ on day $\delta$, 0 otherwise;
- $x_{f,t,d}$—Binary, has value 1 if front $f$ is harvesting stand $t$ on day $d$, with $1 \leq d \leq Tmax$, 0 otherwise;
- $\Delta_{f,t_1,t_2,\delta}$—Binary, has value 1 if front $f$ moved from stand $t_1$ to stand $t_2$ on day $\delta$, 0 otherwise.

Using the data and variables described previously, the following mathematical model was built in order to minimize the harvesting costs, taking into account operating, movement, and transportation costs. Moreover, we ensured that the demand for each product in each demand center was met along the planning horizon.

$$Min \quad Z = \sum_{t \in T} \sum_{p \in P} \sum_{c \in C} \sum_{\delta=1}^{HP} \sum_{f \in F} \left( y_{f,t,\delta} \cdot \theta_{t,f,\delta,p,d} \cdot \pi_{t,c} \cdot \omega \right) +$$
$$\sum_{f \in F} \sum_{t \in T} \sum_{d=1}^{Tmax} \sum_{\delta=1}^{HP} \left( y_{f,t,\delta} \cdot A_{t,f,\delta,d} \cdot \varphi_f \right) + \sum_{f \in F} \sum_{t_1 \in T} \sum_{t_2 \in T} \sum_{\delta=1}^{HP} \left( \Delta_{f,t_1,t_2,\delta} \cdot \rho_{t_1,t_2} \cdot \mu_f \right) \tag{1}$$

subject to:

$$\sum_{f \in F} \sum_{\delta=1}^{HP} y_{f,t,\delta} \leq 1 \quad t \in T \tag{2}$$

$$\sum_{t \in T} y_{f,t,1} = 1 \qquad f \in F \tag{3}$$

$$x_{f,t,1} = y_{f,t,1} \qquad f \in F, \, t \in T \tag{4}$$

$$\sum_{t \in T} x_{f,t,d} \leq 1 \qquad f \in F, \, 1 \leq d \leq Tmax \tag{5}$$

$$\sum_{f \in F} x_{f,t,d} \leq 1 \qquad t \in T, \, 1 \leq d \leq Tmax \tag{6}$$

$$\sum_{t \in T} x_{f,t,d} = 1 \qquad f \in F, \, 1 \leq d \leq HP \tag{7}$$

$$\sum_{t \in T} \sum_{\delta=1}^{HP} \left( y_{f,t,\delta} \cdot \tau_{t,f} \right) \leq HP \qquad f \in F \tag{8}$$

$$\sum_{d=1}^{Tmax} x_{f,t,d} \leq \tau_{t,f} \qquad t \in T, \, f \in F \tag{9}$$

$$\sum_{d=\delta}^{\delta+\tau_{t,f}-1} x_{f,t,d} \geq y_{f,t,\delta} \cdot \tau_{t,f} \qquad \begin{cases} t \in T, f \in F \\ 1 \leq \delta \leq HP \end{cases} \tag{10}$$

$$x_{f,t,d} \geq y_{f,t,\delta} \qquad \begin{cases} f \in F, \, t \in T, \, 1 \leq \delta \leq HP \\ \delta \leq d \leq \left( \delta + \tau_{t,f} - 1 \right) \end{cases} \tag{11}$$

$$\sum_{t \in T} \sum_{f \in F} \sum_{\delta=1}^{HP} \left( y_{f,t,\delta} \cdot \theta_{t,f,\delta,p,d} \right) \geq \lambda_{c,p,\delta} \qquad \begin{cases} p \in P, c \in C \\ 1 \leq d \leq Tmax \end{cases} \tag{12}$$

$$\Delta_{f,t_1,t_2,1} = 0 \qquad f \in F, t_1, t_2 \in T \tag{13}$$

$$\Delta_{f,t_1,t_2,\delta} \geq x_{f,t_2,d} + x_{f,t_1,d-1} - 1 \qquad \begin{cases} f \in F, t_1, t_2 \in T \\ 1 \leq \delta \leq HP \end{cases} \tag{14}$$

$$y_{f,t,\delta} \in \{0, \, 1\} \quad x_{f,t,d} \in \{0, \, 1\} \quad \Delta_{f,t_1,t_2,\delta} \in \{0, \, 1\}$$

The objective function, Equation (1), minimizes the total harvesting cost and is composed by three terms. The first one expresses the transportation cost of the harvested timber to the demand centers. The second term represents the operating cost of the harvesting machinery. Finally, the third term computes the movement cost of the machinery between stands along the planning horizon.

The constraint in Equation (2) guarantees that each stand will only be harvested once along the planning horizon. The constraints in Equations (3) and (4) ensure that each harvesting front starts harvesting a single stand on the first day of the planning horizon. The constraint in Equation (5) prevents a harvesting front from operating in more than one stand in a same day of the PH, i.e., the fronts operate in only one stand per day. The constraint in Equation (6) ensures that stands are harvested by only one harvesting front, i.e., two fronts do not operate simultaneously in a same stand.

The constraint in Equation (7) enforces that harvesting fronts will operate in every day of the planning horizon, and Equation (8) establishes the available operating time span for each harvesting front. The constraints in Equations (9)–(11) guarantee that once a harvesting front starts harvesting a stand, it will operate in this same stand until it is completely harvested.

The constraint in Equation (12) ensures that the amount of product to be transported to the demand centers does not exceed the amount of product harvested on the stands and, simultaneously, meets the demand for each product in the demand centers.

Finally, the constraints represented by inequalities in Equations (13) and (14) activate the variable $\Delta_{f,t_1,t_2,\delta}$, which describes the movement of the cutting fronts between the stands' constraints in Equation (13) and establishes that no movement occurs at the starting day of the harvesting operation in a stand. The constraint in Equation (14) indicates if a harvesting front is moved from stand $t_1$ to stand $t_2$ in a given day of the PH.

*2.2. Case Study*

In order to test our novel approach, we constructed 13 problem instances with increasing levels of complexity (see Table 1). The scenarios tested were randomly generated, varying the number of harvesting fronts, products, and the planning horizon length. All scenarios considered two terrain types and one demand center. The terrain types defined the productivity of the harvesting fronts. Scenario 1 had 12 stands that produced a single assortment, and the terrain was distinguished in types 1 and 2. Two distinct cutting fronts were established for harvesting these stands, such that each front had distinct operating costs, movement costs, and productivities. The planning horizon was defined as 7 days, and the demand center maintained a constant daily demand for a single product.

**Table 1.** Scenarios tested in the analysis. PH stands for the number of days in the planning horizon. A1, A2, and A3 refer to the three different products (assortments) used in the model. T1 and T2 refer to the terrain types (1 and 2), Var. refers to the number of variables, Const. refers to the number of constraints and Tmax refers to the model parameter Tmax.

| Scenario | Mean Area (ha) | PH (Days) | Average Distance to Demand Center (km) | Stand Number (N) | | Stock (m³) | | | Var. (N) | Const. (N) | Tmax (N) |
|---|---|---|---|---|---|---|---|---|---|---|---|
| | | | | T1 | T2 | A1 | A2 | A3 | | | |
| 1 | 17.8 | 7 | 20.2 | 4 | 8 | 1173 | - | - | 4248 | 2962 | 14 |
| 2 | 7.8 | 7 | 15.9 | 7 | 13 | 1463 | 424 | 26 | 19,380 | 10,039 | 16 |
| 3 | 7.8 | 15 | 15.9 | 7 | 13 | 1463 | 424 | 26 | 29,940 | 22,479 | 24 |
| 4 | 9.0 | 15 | 18.1 | 29 | 21 | 3529 | 959 | 51 | 170,700 | 119,521 | 23 |
| 5 | 8.3 | 15 | 24.3 | 22 | 28 | 3406 | 852 | 38 | 178,350 | 119,289 | 24 |
| 6 | 8.6 | 15 | 22.1 | 26 | 24 | 3661 | 1049 | 64 | 178,350 | 119,349 | 24 |
| 7 | 9.8 | 15 | 15.2 | 37 | 13 | 3554 | 1013 | 52 | 178,350 | 119,694 | 24 |
| 8 | 9.3 | 15 | 16.5 | 23 | 27 | 3488 | 913 | 46 | 178,350 | 120,294 | 24 |
| 9 | 9.3 | 15 | 16.2 | 22 | 28 | 3404 | 878 | 43 | 178,350 | 120,309 | 24 |
| 10 | 8.6 | 15 | 29.4 | 31 | 19 | 3573 | 956 | 48 | 178,350 | 118,929 | 24 |
| 11 | 8.8 | 15 | 23.2 | 15 | 35 | 3425 | 879 | 40 | 178,350 | 120,429 | 24 |
| 12 | 8.4 | 15 | 22.9 | 3 | 47 | 3520 | 1004 | 57 | 178,350 | 120,849 | 24 |
| 13 | 9.7 | 15 | 26.3 | 27 | 23 | 3577 | 960 | 51 | 178,350 | 120,234 | 24 |

Scenario 2 included 20 stands that produced up to three distinct forest (1, 2, or 3) products. Three distinct cutting fronts were defined to harvest the stands, such that each front had distinct operating costs, movement costs, and productivities. The planning horizon was defined as 7 days, and the demand center maintained a constant daily demand for each product. Scenario 3 applied the same parameters as scenario 2, but considered a 15-day planning horizon.

Scenarios 4 to 13 had 50 stands available for harvesting that produced up to three distinct forest products. Three distinct cutting fronts were defined to harvest the stands, such that each front had distinct operating costs, movement costs, and productivities. The planning horizon was defined as 15 days, and the demand center maintained a constant daily demand for each product. Moreover, scenario 4 displayed 186,753 variables, with 170,700 of those being integer (binary) variables, and 135,574 were constraints. The remaining scenarios (5 to 13) had 195,003 variables, of which 178,350 were integer (binary) variables, and 135,942 were constraints. Although scenarios 5 to 13 had the same dimensions, the costs, terrain characteristics, and demand-center locations were different, giving rise to different optimal allocations of the harvesting fronts.

We highlight that the scenarios were generated randomly, but based on real data. All parameters and data used in the scenarios were shared by students of the Forest Engineering Graduate School of the Federal University of Paraná. The harvesting productivity parameters, namely $A_{t,f,\delta,d}$ and $\theta_{t,f,\delta,d}$, were assessed based on time and motion studies (e.g., [20]) conducted by the students. In this sense, the scenarios tested do not represent a particular forest area, but demonstrate how the model may be used in diverse forest planning problems for typical conditions of pine and eucalypt forest

plantations in Southern Brazil. To develop the scenarios, we employed the tool developed in [21]. This tool generates forest-planning scenarios based on the total forest area, the topographic characteristics, and the number of demand centers in a given planning problem. The demand centers are randomly distributed inside the area. Subsequently, the specified number of stands is generated within a given radius of the demand centers and respecting minimum and maximum area limits. Finally, the stands are populated with a forest species, topographic conditions, harvesting conditions under high soil moisture, number of forest products, and the productive capacities for each product.

The proposed formulation was implemented by using the Visual Basic programming language on Microsoft Visual Studio Community 2013 software (Microsoft, Redmond, WA, USA)—version 12.0.21005.1 REL (2013). The MILP models were solved with the software application Gurobi (Gurobi Optimization, LLC., Beaverton, OR, USA) —version 8.1.1 (2019), using a computer with an Intel® Core™ i7-4510U CPU @ 2.00 GHz processor and 8 GB of memory.

## 3. Results

We perceived that a larger number of stands in the planning problem led to a substantial increase in the model size and complexity, cascading to the required solution time. This is a key feature of problems involving operational planning, since these need to be performed at a daily or weekly basis. For example, a planning problem involving 20 stands increased the number of variables and constraints in 339% and 456%, compared to a scenario composed by 12 stands. Figure 2 shows the optimization results, in terms of the objective function value and solution time, for all 13 scenarios tested. We noticed that the scenarios with 12 and 20 stands (1 and 2) could be quickly solved, showing processing times below 10 s. Additionally, the scenarios belonging to the same order of magnitude had distinct complexity and computational burden. Scenarios 5 to 13 had a similar number of variables and constraints, but the processing time ranged between 1.3 and 16.0 h.

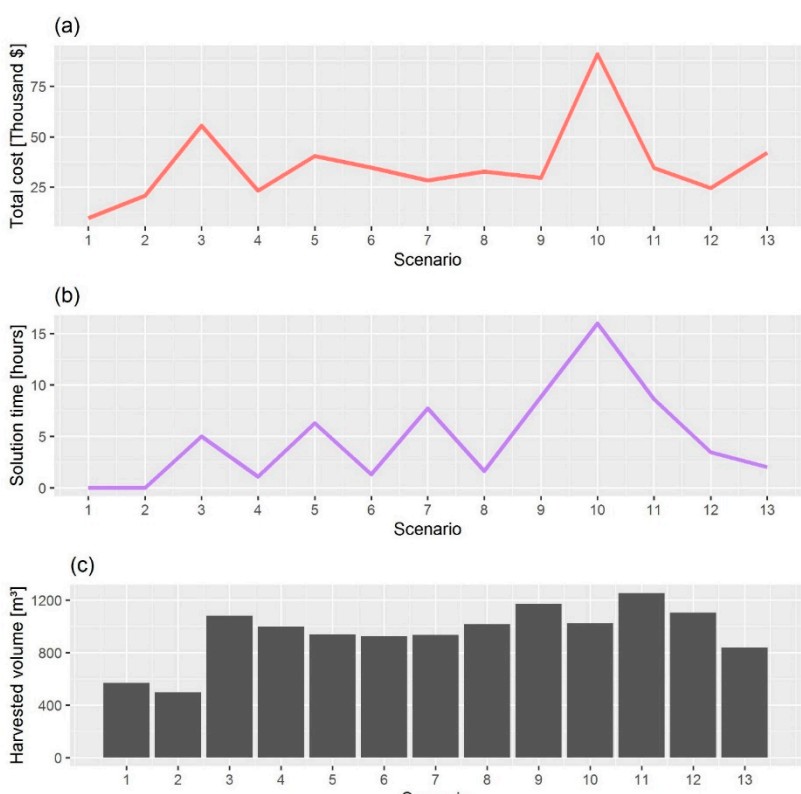

**Figure 2.** Optimization results for the 13 scenarios tested, in terms of the objective function value (**a**), solution time (**b**), and harvested volume (**c**).

The unitary costs of wood supply ranged from 17 to 89 $/m$^3$ in the scenarios tested, representing the variety of operational conditions for forest plantations in Southern Brazil. As expected, the average distance to the demand centers and the proportion of the less favorable terrain types in the scenarios were determinant to unitary production costs. An increase in either of these variables led to an increase in production costs. Additionally, the average stand area affected the solution, with lower costs in scenarios with larger average stand area. Taking into account the results of scenarios 5 to 13, the average distance to the demand centers and the proportion of the less favorable terrain types were capable to explain a large share of the total variation (adjusted $R^2$ = 0.75) in the unitary costs of production. Moreover, we perceived that there was an effect of the harvesting cost on the model solution time ($R^2$ = 0.56), indicating that model instances involving more challenging topographic conditions and larger operational costs become increasingly more complex.

The contribution of the different cost factors to the total cost of harvesting operations for selected scenarios is shown in Table 2. We noted that the transportation cost of forest products from the stands to the demand center was the dominant component of the total harvesting cost, amounting to more than 75% of the total ($30,320.5 and $16,304.8). The operating cost represented 13% of the total cost for scenario 5 and 10% for scenario 2, with a slighter higher contribution than the movement cost, which amounted to 12% and 11% of the total, respectively. Figure 3 illustrates the respective optimal solution for the same scenario, composed by 50 stands and a planning horizon of 15 days. Within this time span, each front completely harvested three stands, with differing operating times. The fronts were scheduled so that the three harvesting fronts were not moved between stands in a same day of the planning horizon, allowing a continuous supply of wood in the demand center. Additionally, the harvesting was prioritized in stands closer to the demand center (see Figure 4), thus reducing the transportation costs related to the demand for different assortments and the total cost of harvesting operations (since this was the dominant cost component). Nevertheless, in the optimal scheduling in scenario 5, we found the displacement of harvesting front 1 to stand 3, at the end of the planning horizon, despite the long movement and transportation distances. This pattern was a result of the combination of favorable topographic conditions and the stock availability in this stand to meet the daily demand for the largest wood assortment.

**Table 2.** Breakdown of the optimal solution for the scenario 5, composed by 50 stands and a 15-day planning period.

| Results | Value ($) | |
| --- | --- | --- |
| | Scenario 2 | Scenario 5 |
| Total cost | 20,770.5 | 40,427.3 |
| Operating Cost | 2180.9 | 5255.5 |
| Cost of Movement | 2284.8 | 4851.3 |
| Cost of Transport | 16,304.8 | 30,320.5 |
| Wood supply | Volume harvested (m$^3$) | |
| | 499.0 | 938.4 |

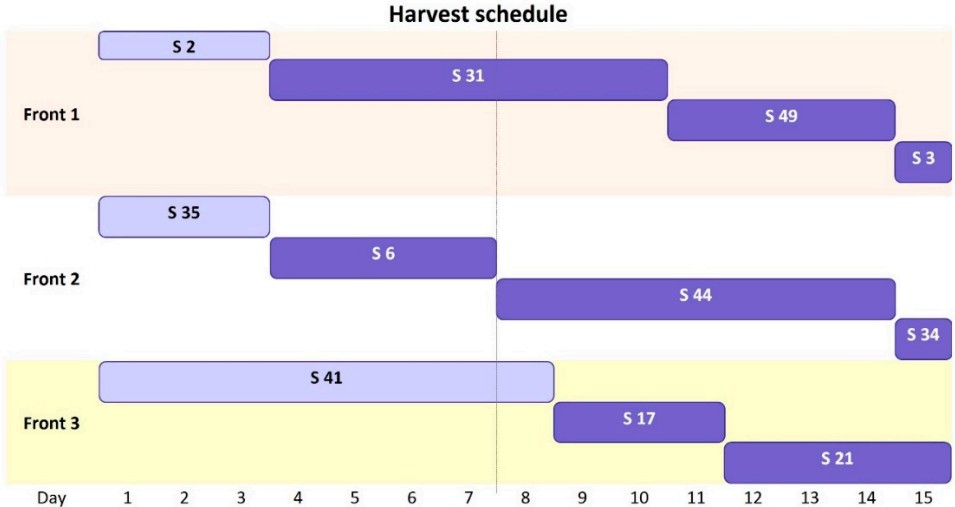

**Figure 3.** Optimal harvesting front scheduling for scenario 5, composed by 50 stands and a 15-day planning period. The sequence of harvested stands is indicated by S and the respective stand number.

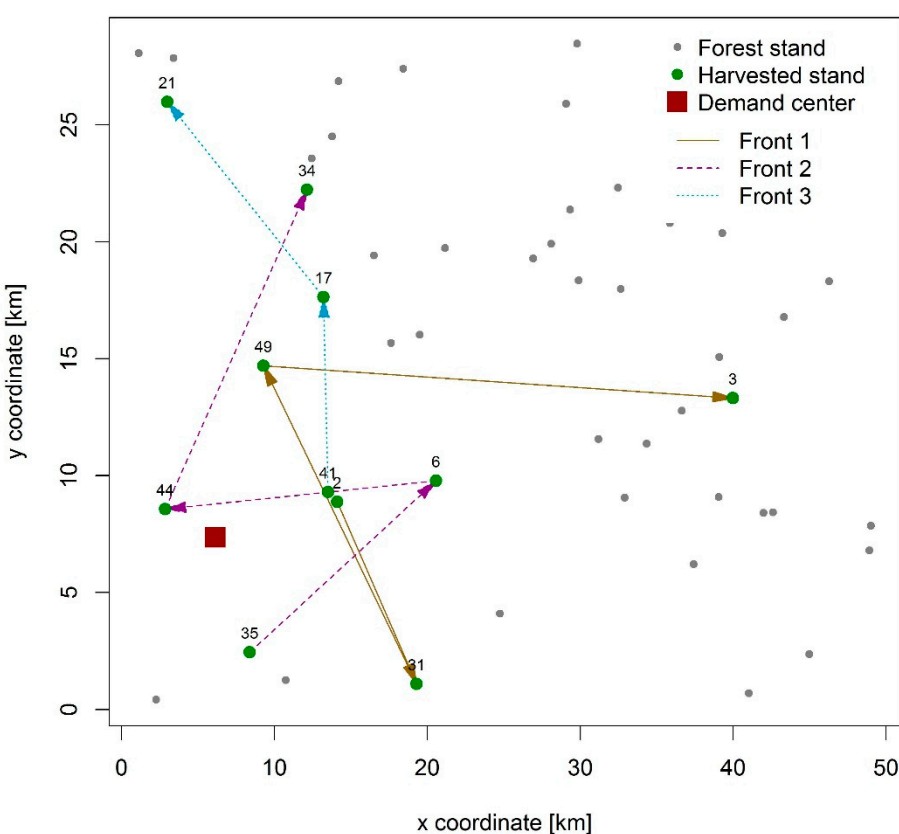

**Figure 4.** Spatial configuration of the optimal solution for scenario 5.

## 4. Discussion

In this paper, we proposed a novel approach for the operational planning of forest harvesting. Our model aimed to minimize the total cost of harvesting activities, including operating, movement, and wood transportation costs, in the face of demand constraints required by demand centers. Our model allows for the optimal allocation of harvesting fronts across forest stands, at each day of the planning horizon, providing valuable information to forest planners, in order to increase the efficiency

of these operations and promote forest profitability. Our results show that the model can be successfully applied to a series of forest conditions. The problem size and the processing time required for the solution, however, substantially grow with the number of stands and the length of the planning horizon considered.

The solution times obtained in the model instances considered here were dependent on the size of the problem generated, with up to a 19 h processing time for the most complex scenario. These values are compatible with forest planning problems at the strategic and tactical levels described in the literature. Könnyű et al. [22] proposed a model for the creation of mature forest patches in an optimized harvest scheduling model and reported solution times ranging from 14 s to 6 h. Augustynczik et al. [1] reported solution times over 3 h for an integrated strategic and tactical forest planning model. Tóth et al. [23] obtained solution times up to 5.4 h in a strategic optimization model with adjacency constraints.

Our results suggest that the dominant cost of forest harvesting and transportation refers to the transportation costs to the demand centers, amounting to 75% of the total cost. We also observed that movement costs had a similar contribution to the operation costs related to the harvesting activities. This is in line with other studies analyzing the supply chain in the forest sector. Ralevic et al. [2] analyzed the operational costs related to the harvest and transport of logging residues for energy production purposes in Canada, and their results show that the transport to the demand centers is the dominant cost factor and represents over 45% of the total cost. Other studies report the transportation costs approaching 30% of the total costs of the wood supply chain (e.g., [24,25]). We can only speculate about the reasons for this behavior, but the higher contribution of transportation costs in our study may be related to the high efficiency of operations in our study area and the characteristics of the terrain and transport distances to the demand center.

Taking into account the substantial impacts of transportation component regarding the total cost of harvesting operations, alternatives to reduce transport and movement distances are required in order to increase the reliability of the supply chain. Our results show that harvesting may be centered in stands closer to the demand centers, whenever the assortment structure of the stands allow for the wood demand to be satisfied and the harvesting takes place within reasonable limits of the optimal thinning or rotation age of individual stands. Moreover, operational and tactical concerns related to harvesting activities need to be further integrated into strategic planning models (e.g., [1,26]). In particular, the aggregation of harvesting activities may exert a beneficial impact on the operational efficiency of harvesting operations and wood transport, since the same roads can be used to transport the production of several stands, and there is less displacement costs of trucks and machinery between harvesting fronts. The cost reduction obtained through clustering harvesting activities, however, needs to be weighed against the total economic loss associated with the formation of harvesting blocks.

We highlight that disruptions in the supply chain may result in large economic impacts on demand centers, due to the high opportunity costs related to processing plants. Harvesting activities and wood transport, however, are subject to a series of uncertainty sources that may disturb the wood supply, such as the climatic conditions, soil conditions, and fluctuations on the demand. Robust and stochastic optimization methods may be applied to cope with sources of uncertainty in the scheduling of harvesting operations [27]. Hence, models may include robustness criteria on the supply chain, such as the time-to-recovery [28] and time-to-survive, in order to create a supply chain that is less susceptible to disruptions.

The proposed mathematical model demonstrated effectiveness and enabled the differentiation of the components of the total cost, so that the work sequence of the cutting fronts and the costs involved in each part of the harvest and the transport of products could be understood. This information becomes crucial, especially considering that the harvesting operations work with tight deadlines and need constant update in a daily and weekly basis, due to uncertainty related to operating conditions (e.g., weather, machinery availability, and workforce availability). Given the flexibility of our model, these conditions can be seamlessly integrated and aid in the decision-making process.

The decision variables of our model were binary in nature, since we aimed to integrate operational aspects of forest harvesting and wood transport, allocating the harvesting machinery in a daily time step. We aimed to develop a model that could be handled with the application of commercial solvers (for the sake of applicability). Hence, we formulated the planning problem as a PILP model for the allocation of harvesting fronts (e.g., [29]). This can lead to a larger model, due to the creation of auxiliary variables to track if fronts are operating in a defined stand in a defined day of the planning horizon. However, PILP binary problems are notably easier to solve. A series of heuristics were proposed, such as binary cuts [30], and built into commercial solver to deal with this class of problems. Taking into account the large number of constraints and binary variables generated when the planning problem involves a large number of stands, heuristic solution methods may be required. These approaches can be applied to obtain reasonable solutions in a shorter processing time and deserve further investigation. Heuristic solution options may include, for example, simulated annealing and genetic algorithms, which have been applied in the past decades to a variety of forest strategic and tactical forest planning problems (e.g., [31–34]). Heuristic solutions can also be applied as a starting point for exact methods, which may improve the solution obtained. This may be particularly useful if exact methods require a long processing time to find good feasible solutions.

We included here demand constraints and the operational costs related to the harvesting activities. Although the costs of harvesting operations are the main driver of decision-making, sustainability aspects of forest harvesting have been increasingly receiving attention in the literature [35–37]. Therefore, extensions to our model may be considered in future studies, e.g., by integrating constraints related to soil protection (considering the water content in the soil and sensibility to compaction) and the emissions related to harvesting and transport operations into the planning model.

## 5. Conclusions

Operational aspects of forest harvesting are determinant to forest profitability and essential to the maintenance of wood supply chains. The planning process of these operations, however, are inherently complex and involve a large number of decisions, in the face of resource limitations (e.g., the demand of production centers, number of available machines, and budget). Optimization methods can be used to successfully tackle such decision problems and aid decision-making regarding the schedule of forestry machinery across stands. The model developed here can be applied to allocate harvesting fronts to minimize harvesting and transportation costs in a variety of forest conditions; thereby, managers can better plan forest operations and efficiently achieve their goals.

**Author Contributions:** Conceptualization, P.A.V.H.d.S. and A.C.L.d.S.; methodology, P.A.V.H.d.S., A.C.L.d.S. and J.E.A.; software, P.A.V.H.d.S. and A.C.L.d.S; validation, P.A.V.H.d.S, A.C.L.d.S., J.E.A. and A.L.D.A.; formal analysis, P.A.V.H.d.S., A.C.L.d.S., J.E.A. and A.L.D.A.; investigation, P.A.V.H.d.S. and A.C.L.d.S.; resources, A.C.L.d.S and J.E.A.; writing—original draft preparation, P.A.V.H.d.S. and A.L.D.A.; writing—review and editing, P.A.V.H.d.S. and A.L.D.A.; supervision, A.C.L.d.S. and J.E.A.

**Funding:** The article processing charge was funded by the University of Freiburg in the funding programme Open Access Publishing.

**Acknowledgments:** This research benefited from the exchange platform provided by the SuFoRun project (Marie Sklodowska Curie Grant Agreement No. 691149).

**Conflicts of Interest:** The authors declare no conflicts of interest.

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
