# Peer review of "A Mathematical Model for the Integrated Optimization of Harvest and Transport Scheduling of Forest Products"

_forests, doi:10.3390/f10121110_

Round 1

Reviewer 1 Report

The article presents the analysis, but does not provide recommendations on the results of studies. Transport planning recommendations for the process chains in question should be displayed.

Author Response

Reviewer #1

The article presents the analysis, but does not provide recommendations on the results of studies. Transport planning recommendations for the process chains in question should be displayed.

Answer: We apologize for the missing recommendations. Our model’s results show that transport to demand center are dominant in the total cost of harvesting operations. In this sense, harvesting may be concentrated in stands closer to demand centers whenever the assortment structure of the stands enable to cover the demand of different products and the stand age is within a reasonable range of optimality, in terms of the ideal age for harvesting. Strategic forest planning models need to be further integrated with tactical and operational aspects of harvesting operations. Particularly, the aggregation of harvesting activities may exert a beneficial impact on the operational efficiency of harvesting operations and wood transport, since the same roads can be used to transport the production of several stands and there is less displacement costs of trucks and machinery between harvesting fronts. The costs, however, need to be compared with the total NPV loss on the long term, by harvesting stands in periods outside the range of optimality, in order to aggregate forest harvesting.  We have now added this discussion in lines 318-330. This has clearly contributed to the quality of our manuscript.

Reviewer 2 Report

General remarks:

The manuscript deal with interesting problem - optimization in scheduling stands for final cut using linear programming. The novelty of research problem is the integration of harvest and transport in operational planning. The proposed model has sufficient description but the description of “Case study” and  “Results should be improved.

Specific comments:

Line 43 – should be “Weintraub and Cholaky [10]…”  instead of:” [10] classify …”.,

Line 44 should be “According to Mitchell [11]” this remark applies also whole manuscript.

Line 56 instead of “a low computational time…” shoud be “...short computational time…”

Line 58 should be “According to Rönnqvist et all [15] …”

Line 70 Give a full name before the abbreviation; when it appears for the first time it should be: “Mixed Integer Linear Problem (MILP) …”

Line 76-77 – the same information was given in lines 50-52.

Line 99 – different forest products mean different timber assortments or also include nontimber products?

Line 131 what “dl” means? – “d” is not defined up to this line

Line 139 is it correct “… front f would start…” or it should be “..front f would finish …”- please check it.

Line 146 in line 127 “t” was defined as a stand and here means a plot but a plot can be a part of stand this remark applies also for the next lines: 150, 151, 153, 155, 156.

158 instead of “..travelled…” maybe it will be better “Total distance of front f movement”.

 Line 164  - F was defined in line 119 as  “Set of cutting fronts” and here means “Total costs”

General remarks for all variables units should be defined: ha. km, m3 $/ha, $/ km … and so on.

Line 192 – 225 Case study – it should be improved. Is it really 13 different scenarios or only “a randomly generated 13 problems”. Each scenario should be described. What is the difference between scenarios from 5 to 13. In the results scenarios should be compared and after that an assessment should be given which solution is better. The total cost is known for each scenario but we do not know which is the total amount of “forest product” harvested and transported to”demand center” ( which is the harvested area) in particular scenarios – it makes it difficult or impossible to compare efficiency between scenarios.

Line 225 – table 1- Terrain types and Demand center are the same in each scenario so this information can be given in the general description, and not in each row of table.

Line 251 – the description of Figure 2 should be improved – what means ‘’bars’’ and what ‘’line’’.

Vertical axis presented total costs in dollars but for which amount of ”forest products” ?

Line 266 – Table 2 – Comparison of costs and efficiency between scenario2 and scenario 5 is impossible without the amount of harvested timber in each scenario.

Author Response

Reviewer #2

General remarks:

The manuscript deal with interesting problem - optimization in scheduling stands for final cut using linear programming. The novelty of research problem is the integration of harvest and transport in operational planning. The proposed model has sufficient description but the description of “Case study” and  “Results should be improved.

Answer: We are most thankful for the interest in our study. We agree that the description of the case study and the results are wanting. We have now restructured these sections, describing the optimization scenarios in more detail and adding the harvesting volume in the results section for each instance solved. This has clearly contributed to the quality and readability of our manuscript.

Specific comments:

Line 43 – should be “Weintraub and Cholaky [10]…”  instead of:” [10] classify …”.,

Answer: we thank the remark and have corrected the text accordingly.

Line 44 should be “According to Mitchell [11]” this remark applies also whole manuscript.

Answer: Answer: we thank the remark and have corrected the text accordingly.

Line 56 instead of “a low computational time…” shoud be “...short computational time…”

Answer: we thank the suggestion and have corrected the text accordingly.

Line 58 should be “According to Rönnqvist et all [15] …”

Answer: we thank the remark and have corrected the text accordingly.

Line 70 Give a full name before the abbreviation; when it appears for the first time it should be: “Mixed Integer Linear Problem (MILP) …”

Answer: we thank the remark and have corrected the text accordingly.

Line 76-77 – the same information was given in lines 50-52.

Answer: we thank the remark and have removed these lines in the new version of the manuscript.

Line 99 – different forest products mean different timber assortments or also include nontimber products?

Answer: That is correct, we refer here to different timber assortments and not to non-timber forest products. We have clarified this in line 100.

Line 131 what “dl” means? – “d” is not defined up to this line

Answer: we apologize for the lack of information here. “dl” refers to the day that is used as indicator for the movement of harvesting fronts between 2 stands. To improve the model’s clarity, we refer to this now as “d” and “d-1”. Moreover, we shifted the order of description of parameters d and .

Line 139 is it correct “… front f would start…” or it should be “..front f would finish …”- please check it.

Answer: We apologize for the confusing definition of the parameter Tmax. In fact, this refers to the last day where a front f could harvest a stand, not necessarily harvesting the whole area. We have now reframed the description of this parameter in lines 140-142.

Line 146 in line 127 “t” was defined as a stand and here means a plot but a plot can be a part of stand this remark applies also for the next lines: 150, 151, 153, 155, 156.

Answer: we apologize for the inconsistency. The plot here refers to stands. We have now replaced the term “plot” by “stand” throughout the manuscript.

158 instead of “..travelled…” maybe it will be better “Total distance of front f movement”.

Answer: we thank the suggestion and have corrected the sentence accordingly.

 Line 164  - F was defined in line 119 as  “Set of cutting fronts” and here means “Total costs”

Answer: we apologize for the misuse of the indicator here. F refers here to the objective function value. To avoid confusion between this and the set of cutting fronts we replaced it by “Min Z=” (line 167).

General remarks for all variables units should be defined: ha. km, m3 $/ha, $/ km … and so on.

Answer: we apologize for the missing information. We have now added the units to the variables in the model description.

Line 192 – 225 Case study – it should be improved. Is it really 13 different scenarios or only “a randomly generated 13 problems”. Each scenario should be described. What is the difference between scenarios from 5 to 13. In the results scenarios should be compared and after that an assessment should be given which solution is better. The total cost is known for each scenario but we do not know which is the total amount of “forest product” harvested and transported to”demand center” ( which is the harvested area) in particular scenarios – it makes it difficult or impossible to compare efficiency between scenarios.

Answer: That is correct, particularly for scenarios 5-13, these were only different randomly generated instances of the problem. Nevertheless, they had different terrain characteristics, standing volumes and location of the demand center. Although, in terms of complexity, i.e. number of constraints and variables, they were identical. We believe that due to the variation in costs and topography, these should be treated as different scenarios. We point to this in lines 215-217. We fully agree that the amount of forest products transported needs to be included in the analysis. We have now added this information in figure2, table 2 and discussed the results in lines 248-256.

Line 225 – table 1- Terrain types and Demand center are the same in each scenario so this information can be given in the general description, and not in each row of table.

Answer: we agree and have removed these columns from the table, adding this information in Table 1.

Line 251 – the description of Figure 2 should be improved – what means ‘’bars’’ and what ‘’line’’.

Answer: we apologize for the lack of clarity in the figure. We have now restructured it to clarify the meaning of each graph (line 283).

Vertical axis presented total costs in dollars but for which amount of ”forest products” ?

Answer: we apologize for the missing information. We have now added the amount of harvested products in each scenario in Figure 2.

Line 266 – Table 2 – Comparison of costs and efficiency between scenario2 and scenario 5 is impossible without the amount of harvested timber in each scenario.

Answer: we fully agree that the harvesting amount need to be included for comparison. This information was added in Table 3 in the new version of the manuscript. Please note, however, that our aim here was to compare the components of the total cost within each scenario, under different problem sizes, rather than compare the efficiency of the two scenarios.

Reviewer 3 Report

The main concern with this paper is related to the mathematical formulation of the problem and its numerical resolution. In my opinion, the model (1)-(18) is unnecessarily complicated. For example, equations (12), (13), (14) and (18) are not authentic constraints. In fact, these equations are showing how to compute the continuous variables from the binary variables. These constraints have to be deleted, and the continuous variables have to be also deleted from the decision variables set. Consequently, the problem is not a MILP but an ILP (in fact a BLP) and it is numerical solution is much easier.

Once the continuous variables are deleted, the possibility of work with integer (not binary) decision variables should be investigated, in order to simplify the model and reduce the dimension of the problem (and, consequently, solve the problem in complex scenarios with a reasonable computation time). Then, a numerical method which gives good results with reasonable computation times should be investigated.

In summary, the mathematical model and how to solve it must be reviewed deeply.

Other comments are the following:

The concept of T_max is not clear. A detailed explanation of this concept is necessary. The differences between δ (a day in Planning Horizon) and d (a harvest day) should be also clarified. Units for some unidimensional parameters (Planning horizon, cost of transport for the forest products), input data and decision variables should be included. Some input data (for example, A or Theta) don’t seem easy to obtain. An explanation as how they are obtained must be given.

Author Response

Reviewer #3

The main concern with this paper is related to the mathematical formulation of the problem and its numerical resolution. In my opinion, the model (1)-(18) is unnecessarily complicated. For example, equations (12), (13), (14) and (18) are not authentic constraints. In fact, these equations are showing how to compute the continuous variables from the binary variables. These constraints have to be deleted, and the continuous variables have to be also deleted from the decision variables set. Consequently, the problem is not a MILP but an ILP (in fact a BLP) and it is numerical solution is much easier.

Answer: We appreciate the discussion points regarding our formulation. We recognize that the formulation of our model may be bulky, due to the large number of auxiliary variables. Nevertheless, we believe our problem was binary in nature, since we wished to allocate the machinery and transport in a daily time step. As properly stated, the solution of models of binary variables is easier, due to the availability of heuristic methods to provide good starting points already built in commercial solvers. Here we had the aim to propose a model that could be directly handled by such solvers (for the sake of applicability). We agree that Eq.12, 13, 14 and 18 are not traditional constraints but rather the computation of auxiliary variables. Nevertheless, the value of these auxiliary variables need to be included as constraints, since their computation is endogenous to the model. Hence, we believe it is necessary to maintain them for the sake of interpretability of the model and for practical implementation purposes.

Once the continuous variables are deleted, the possibility of work with integer (not binary) decision variables should be investigated, in order to simplify the model and reduce the dimension of the problem (and, consequently, solve the problem in complex scenarios with a reasonable computation time). Then, a numerical method which gives good results with reasonable computation times should be investigated.

In summary, the mathematical model and how to solve it must be reviewed deeply.

Answer: We fully agree that a more compact version of the model is worth investigating. However, the main aims of our paper were not only to propose a model for the operational planning that can be readily solved by commercial solvers, but also investigate the relationship between costs as well.  As correctly noticed, a MBLP formulation can take advantage of Binary cuts and other heuristics already implemented in commercial solvers. In this sense, we believe a MILP reformulation could lead to a more compact problem but with longer processing time. We highlight, however, that more compact formulations and exact and heuristics approaches to solve operational planning problem deserves further investigation (lines 348-363).

Other comments are the following:

The concept of T_max is not clear. A detailed explanation of this concept is necessary. The differences between δ (a day in Planning Horizon) and d (a harvest day) should be also clarified. Units for some unidimensional parameters (Planning horizon, cost of transport for the forest products), input data and decision variables should be included. Some input data (for example, A or Theta) don’t seem easy to obtain. An explanation as how they are obtained must be given.

Answer: we apologize for the confusing description of the parameters. T_max refers to the last day a front can start harvesting a stand (not necessarily completing the harvest of this stand), taking into account the time needed to move the harvesting front between stands. The difference between δ (a day in Planning Horizon) and d (a harvest day), is that the first is associated with the y variables and keeps track of the day a harvesting front starts the harvesting of a stand (δ). The latter is associated with the x variables, which indicates if a harvesting front is operating in a defined stand in day d. Hence, they define a starting day and an operating day in the PH. In the new version of the manuscript, we have added this information, the units of the parameters and the sources of input data (lines 125-167). This has clearly contributed to the quality and readability of our manuscript.

Round 2

Reviewer 2 Report

The authors took into account the comments presented in the first review as well as others reviews and made corrections to the manuscript. They gave comprehensive answers regarding comments.

It is currently an interesting work,  and the scientific value is at an appropriate level and is sufficient and coincides with the theme of the Forests magazine.

Remark - in Figure 4 axis should be better described or symbols "x" and "y" explained 

Author Response

We are most grateful for the valuable comments and thorough revision provided by Reviewer #2 and Reviewer #3. We have now performed major revisions in our manuscript. We have fully restructured the optimization model, removing the auxiliary variables and integrating them into the objective function. We have removed the constraints related to the computation of these variables and re-optimized all scenarios with the more compact version of the model. Moreover, we clarify how the input parameters were obtained and revised Figure 4 to explain the axes values. These changes have clearly contributed to the quality of our analysis. We believe the paper now fully complies with the high standard of Forests.

Reviewer #2

The authors took into account the comments presented in the first review as well as others reviews and made corrections to the manuscript. They gave comprehensive answers regarding comments.

It is currently an interesting work,  and the scientific value is at an appropriate level and is sufficient and coincides with the theme of the Forests magazine.

Remark - in Figure 4 axis should be better described or symbols "x" and "y" explained

Answer: we appreciate the interest in our manuscript. We have now revised Figure 4, detailing the x and y axes a spatial coordinates.

Reviewer 3 Report

1.- I appreciate the authors' explanations on the mathematical formulation of the problem (first point of my previous report), but unfortunately I don’t agree with them. Next, I will be more precise to clarify my concerns with the mathematical model:

1.1.- Equation (12): Because ??,?,?,?,? are data, ??,?,?,? are computed directly from the binary variables ??,?,?  and they should not be considered decision variables. Consequently, Eq. (12) is not a constraint but it is only the definition of the auxiliary variables ??,?,?,?.

1.2.- Equation (13): Because A?,?,?,? are data, ??,?,? are computed directly from the binary variables ??,?,?  and they should not be considered decision variables. Consequently, Eq. (13) is not a constraint but It is only the definition of the auxiliary variables ??,?,?.

3.- Equation (14): I don’t understand well this equation. Is it an equality or it should be an inequality? Where should belong d? This equation should be revised, but if it is correct, ??,?,?,?  can be computed directly from ??,?,?,? and they should not be considered decision variables. In that case, Eq. (14) is not a constraint and but is only the definition of the auxiliary variables ??,?,?,? .

4.- Equation (18): Because ??1,?2 are data, ?? are computed directly from the binary variables Δ?,?1,?2,? and they should not be considered decision variables. Consequently, Eq. (18) is not a constraint but it is only the definition of the auxiliary variables ??.

5.- Equations (5) and (6) are redundant for some values of d and, consequently, no necessary.

In short, the mathematical model has to be thoroughly revised: auxiliary variables have to be erased from the decision variables and all not necessary constraints have to be deleted from the model. Consequently, numerical experiments have to be realized and rewritten again: the number of variables and constraints are now much lower and computational times are also expected lower.

2.- In the numerical experiments, the value of Tmax have to be included for different Scenarios. Why the number of variables for Scenario 4 is lower than for Scenarios 5-13? Is it causes by the value of Tmax?

3.- In my previous report I commented “Some input data (for example, A or Theta) don’t seem easy to obtain. An explanation as how they are obtained must be given.” The authors forgot to do it, and it is very interesting to know the usefulness of the proposed model.

Author Response

We are most grateful for the valuable comments and thorough revision provided by Reviewer #2 and Reviewer #3. We have now performed major revisions in our manuscript. We have fully restructured the optimization model, removing the auxiliary variables and integrating them into the objective function. We have removed the constraints related to the computation of these variables and re-optimized all scenarios with the more compact version of the model. Moreover, we clarify how the input parameters were obtained and revised Figure 4 to explain the axes values. These changes have clearly contributed to the quality of our analysis. We believe the paper now fully complies with the high standard of Forests.

Reviewer #3

1.- I appreciate the authors' explanations on the mathematical formulation of the problem (first point of my previous report), but unfortunately I don’t agree with them. Next, I will be more precise to clarify my concerns with the mathematical model:

1.1.- Equation (12): Because ??,?,?,?,? are data, ??,?,?,? are computed directly from the binary variables ??,?,?  and they should not be considered decision variables. Consequently, Eq. (12) is not a constraint but it is only the definition of the auxiliary variables ,,,.

1.2.- Equation (13): Because A?,?,?,? are data, ??,?,? are computed directly from the binary variables ??,?,?  and they should not be considered decision variables. Consequently, Eq. (13) is not a constraint but It is only the definition of the auxiliary variables ,,.

3.- Equation (14): I don’t understand well this equation. Is it an equality or it should be an inequality? Where should belong d? This equation should be revised, but if it is correct, ??,?,?,?  can be computed directly from ??,?,?,? and they should not be considered decision variables. In that case, Eq. (14) is not a constraint and but is only the definition of the auxiliary variables ??,,, .

4.- Equation (18): Because ??1,?2 are data, ?? are computed directly from the binary variables Δ?,?1,?2,? and they should not be considered decision variables. Consequently, Eq. (18) is not a constraint but it is only the definition of the auxiliary variables ??.

Answer: we are most thankful for pointing these improvements to our model. This is absolutely correct, constraints 12 to 14 and 18 may be removed from the model. We have now fully revised and re-optimized all scenarios, removing all the constraints related to the auxiliary variables (Eq. 12, Eq. 13, Eq. 14 and Eq. 18) and integrating them directly into the objective function (section 2.1). In the new version of the model, there was an approximate 12% reduction in the solution time (ranging from 1.6 to 50%). This has clearly improved our analysis.

5.- Equations (5) and (6) are redundant for some values of d and, consequently, no necessary.

Answer: In fact, these two constraints are not redundant. Constraint  Eq. (5) ensures that a front operates in only one stand in each day of the PH, whereas constraint Eq. (6) prevents that two harvesting fronts to operate in a same stand simultaneously. For example, consider the case where there are 2 harvesting fronts, 3 stands and 1 day in the PH. These constraints would read:

Eq(5):

x_111 + x_121 + x_131 ≤ 1

x_211 + x_221 + x_231 ≤ 1

Eq(6):

x_111 + x_211 ≤ 1

x_121 + x_221 ≤ 1

x_131 + x_231 ≤ 1

We may see that both constraints are required to maintain only one front operating in one stand in day in the PH. Moreover, we have performed tests alternately removing constraints Eq. (5) and Eq. (6) from the model and we verified that the solutions violated the assumptions of our model.

We also would like to highlight that the solver applied detects automatically redundancies in the constraints and removes them in the pre-solve phase (Achterberg et al. 2019).

In short, the mathematical model has to be thoroughly revised: auxiliary variables have to be erased from the decision variables and all not necessary constraints have to be deleted from the model. Consequently, numerical experiments have to be realized and rewritten again: the number of variables and constraints are now much lower and computational times are also expected lower.

Answer: We fully agree and have now removed all the auxiliary variables from the model and re-optimized all scenarios with this more compact formulation (section 2.1 and table 1). We are most thankful for the suggestion. This has clearly contributed to the quality of our analysis.

 2.- In the numerical experiments, the value of Tmax have to be included for different Scenarios. Why the number of variables for Scenario 4 is lower than for Scenarios 5-13? Is it causes by the value of Tmax?

Answer: That is correct, the Tmax in scenario 4 was different than the ones in 5 to 13. We have now included the value of Tmax in the table describing the scenarios (Table 1).

3.- In my previous report I commented “Some input data (for example, A or Theta) don’t seem easy to obtain. An explanation as how they are obtained must be given.” The authors forgot to do it, and it is very interesting to know the usefulness of the proposed model.

Answer: We are deeply sorry that we overlooked this comment in our previous response. These productivity values (A and theta) were actually measured in the field by the students of the university, through a time and motion study (e.g. Suchomel et al. 2011). Based on the productivity of harvesting fronts in different terrains and the time of a daily shift, we were able to derive the A_(t,f,δ,d) e θ_(t,f,δ,p,d) values. We added this information in lines 217-220 and 150-157.

Suchomel, C., Becker, G., & Pyttel, P. (2011). Fully mechanized harvesting in aged oak coppice stands. Forest Products Journal, 61(4), 290-296. Achterberg, T., Bixby, R. E., Gu, Z., Rothberg, E., & Weninger, D. (2019). Presolve reductions in mixed integer programming. INFORMS Journal on Computing.
